# The Experience of Women Giving Birth after Cesarean Section—A Longitudinal Observational Study

**DOI:** 10.3390/healthcare11121806

**Published:** 2023-06-20

**Authors:** Dorota Sys, Anna Kajdy, Martyna Niżniowska, Barbara Baranowska, Dorota Raczkiewicz, Urszula Tataj-Puzyna

**Affiliations:** 1Department of Medical Statistics, School of Public Health, Centre of Postgraduate Medical Education, 01-826 Warsaw, Poland; dorota.raczkiewicz@cmkp.edu.pl; 2First Department of Obstetrics and Gynecology, Centre of Postgraduate Medical Education, 01-004 Warsaw, Poland; akajdy@cmkp.edu.pl; 3Saint Sophia Specialist Hospital, 01-004 Warsaw, Poland; 4Department of Midwifery, Centre of Postgraduate Medical Education, 01-004 Warsaw, Poland; barbara.baranowska@cmkp.edu.pl (B.B.); urszula.tataj-puzyna@cmkp.edu.pl (U.T.-P.)

**Keywords:** parturition, natural childbirth, trial of labor, labor after cesarean section, vaginal birth after cesarean, cesarean section

## Abstract

Natural childbirth after a previous cesarean section is a debated issue despite scientific research and international recommendations. This study aimed to examine the experiences of women giving birth after a previous cesarean section, their preferences, and changes in attitudes towards childbirth after labor. This longitudinal study involved 288 pregnant women who had a previous cesarean section and completed a web-based questionnaire before and after labor, including information about their obstetric history, birth beliefs, and preferred mode of delivery. Among women who preferred a vaginal birth, nearly 80% tried it and 49.78% finished delivery by this mode. Among women declaring a preference for an elective cesarean section, 30% attempted a vaginal birth. Choosing a hospital where staff supported their decision (regardless of the decision) was the most helpful factor in preparing for labor after a cesarean section (63.19%). Women’s birth preferences changed after labor, with women who had a vaginal birth after a cesarean section preferring this mode of delivery in their next pregnancy (89.34%). The mode of birth did not always follow the women’s preferences, with some women who preferred a natural childbirth undergoing an elective cesarean section for medical reasons. A variety of changes were noticeable among women giving birth after a cesarean section, with a large proportion preferring natural birth in their next pregnancy. Hospitals should support women’s birth preferences after a cesarean section (if medically appropriate), providing comprehensive counseling, resources, and emotional support to ensure informed decisions and positive birth experiences.

## 1. Introduction

Scientific associations from the United Kingdom (Royal College of Obstetricians and Gynaecologists), the United States (The American College of Obstetrics and Gynecology), and Canada (Society of Obstetricians and Gynecology of Canada) have published recommendations on childbirth after a cesarean section (CS). All associations agree that natural childbirth should be considered after one or two cesarean sections. All of the documents include information on detailed medical management, as well as recommendations on how to collaborate with pregnant woman in determining each birth method. Both the opportunities and risks of each method should be presented comprehensively, in clear language, and at each stage of pregnancy [1,2,3].

Following the above-mentioned documents, national societies including those in France and Austria have issued recommendations in this regard [4,5]. The Polish Society of Gynecologists and Obstetricians has also expressed a statement on this issue several times. Although there has not been a document dedicated completely to childbirth after a cesarean section, the 2018 recommendations on cesarean sections include those on childbirth after a previous cesarean section [6].

It is possible to achieve natural childbirth after a cesarean section. Several studies and reviews show that natural childbirth is safe for the mother and baby when the parturient has no other contraindications and risk factors [7]. Studies show that there is a 74% chance of a successful vaginal birth, so it should be the first-line choice in the absence of additional contraindications [7,8]. However, the decision should be made on a case-by-case basis depending on the obstetric history, current conditions, examination results, and risk assessment, as well as the woman’s preference. Despite scientific research and international and national recommendations, experience shows that natural childbirth after a previous cesarean section is a debatable issue among both medical personnel and women.

The advantages of a vaginal delivery over a C-section are multifold and significant. Mothers who undergo a vaginal birth have shown quicker recovery times, fewer hospital readmissions, and lower postpartum depression rates [9]. They also face lower risk of future reproductive complications, such as placental issues and uterine rupture, in subsequent pregnancies [10]. For the newborn, the birth canal’s passage stimulates the initiation of beneficial gut microbiota, aiding in the development of the baby’s immune system [11]. Moreover, a study by Sevelsted et al. confirmed that a vaginal birth reduces the likelihood of various neonatal respiratory issues, such as transient tachypnea, which are more prevalent in babies delivered through C-section [12]. Recognizing these benefits can contribute to informed decision-making among expectant parents and healthcare providers. The World Health Organization, in recommendations issued in 2018, highlighted that a positive childbirth experience includes both obstetric outcomes and the woman’s experience. A positive birth experience is determined by considering the woman’s perspective and expectations based on their personal beliefs and opinions. In addition, systematic reviews show that perinatal outcomes are driven not only by the level of medical knowledge, routine examinations, and procedures applied, but also largely by holistic high-quality care, which should be focused on the woman and her needs [13].

In this regard, it is important to recognize the needs and expectations of women about childbirth after a cesarean section. An important aspect in this context is the possibility of the woman’s participation in decision-making regarding the procedures performed. In the case of childbirth by cesarean section, medical indications can significantly limit the possibility of a woman’s co-decision and even exclude the birth process from the woman’s expectations and needs. However, leaving open the possibility of a woman’s co-decision-making and considering her perspective even when specific medical procedures need to be performed may give a woman a sense of control over the childbirth process and thus a positive childbirth experience [14].

Exploring women’s perspectives on childbirth after a cesarean section is not only cognitively interesting but also important from the point of view of all those involved in the decision-making process about the mode of birth. Longitudinal cohort studies will make it possible to identify patterns occurring in this regard and recognize significant variations depending on women’s preferences and the labor process.

This study aimed to examine the experiences of women giving birth after a cesarean section, their preferences, and changes in attitudes towards childbirth after labor. The secondary goals were to determine the correlations between women’s preferences and the process of childbirth and to verify changes in women’s attitudes towards childbirth after labor after a cesarean delivery.

## 2. Materials and Methods

### 2.1. Design

A longitudinal cohort survey was conducted using the CAWI (Computer Assisted Web Interview) technique. Data were collected using online questionnaires, which allowed us to collect data from a large group of respondents while maintaining anonymity and the comfort of study participants, who could complete the questionnaire anywhere and anytime using their computers or mobile devices. Participants were surveyed at two time points: during pregnancy and between six weeks and six months after childbirth. STROBE and CHERRIES were used for reporting this study [15,16].

### 2.2. Setting

The study was conducted in Poland while The Polish Society of Gynecologists and Obstetricians’ recommendations allowing women to make a co-decision about the mode of labor after a cesarean section were in effect. In the case of a physician’s recommendation to try vaginal delivery, the woman had the option not to give consent. As a result, an elective cesarean section was performed [17].

### 2.3. Study Group

The study group included pregnant women who had previously undergone a minimum of one cesarean section birth. The inclusion criteria for the study were: women who were currently pregnant, had a previous cesarean section, were age at least 18 years, and provided consent for the study. The exclusion criterion was an indication for cesarean section in the current pregnancy.

Before completing the first questionnaire, the study participants were informed about the purpose of the study and the use of the data for scientific purposes only. The questionnaires were anonymous. Those willing to participate were invited to leave their email address at the end of the questionnaire to participate in the second stage of the study. In the beginning, the participant was informed that completing the questionnaire was tantamount to consenting to participate in the study. Participants did not receive any form of compensation for participating in the study.

### 2.4. Sample Size

The population of pregnant women after a previous cesarean section in Poland is not precisely known. Statistics in this regard are not kept by either the Central Statistical Office or the National Health Insurance Agency. The population was estimated based on the number of births of second and subsequent children in 2017 (*n* = 229,326) [18] and the average rate of cesarean sections in Poland (42.2%) [19]. Considering the aforementioned data, the estimated population of pregnant women after a previous cesarean section was 96,776. The sample size, assuming a confidence level of 95%, a maximum error of 5%, and a proportion of 42%, was 375 women (calculations were made using the website https://www.calculator.net/sample-size-calculator.html accessed on 6 December 2021).

### 2.5. Tool

The research tool consisted of the authors’ questionnaire in two parts. The questionnaire was created by an interdisciplinary team of experts that included midwives, obstetrician-gynecologists, and a sociologist based on a review of the literature.

The first part of the questionnaire, completed during pregnancy, contained 30 closed-ended single- and multiple-choice questions, including four matrix questions. The questions concerned the obstetric history, opinions, and attitudes toward future childbirth after cesarean section, including preferences and motivations, as well as concerns about the upcoming childbirth. The second part of the questionnaire, completed after labor, contained 20 questions, including three matrix questions and two open-ended questions. The questions focused on the process of labor, including attempting natural childbirth. (see Appendix A) Validation of the questionnaire consisted of two stages: the first was a pilot study, the second was evaluation by the researchers, followed by implementation of the corrections. In the pilot study, the questionnaire was distributed to pregnant women asking them to fill out the questionnaire and provide their comments, concerns, or describe any difficulties in filling it out, if any. The comments were then analyzed by the research team and those that were applicable were incorporated. In addition, we conducted a reliability analysis for the scale used in order to identify the factors influencing the mode of delivery. Cronbach’s alpha for this scale was 0.821.

### 2.6. Recruitment

The questionnaire was entered into an online platform (https://interaktywnie.com/ accessed 12 June 2023) that allowed the questionnaire to be shared online. Once the questionnaire was entered into the system, a link to each of the two parts of the questionnaire was generated allowing access to the survey.

Recruitment of pregnant women for the study was carried out through social media and parenting portals. A link with a description of the study was posted in open and closed groups and forums dedicated to pregnant and postpartum women. (see Appendix A) The scheme for recruiting participants for the study is shown in the flow charts in Figure 1 and Figure 2.

### 2.7. Measures

The study measured a range of variables related to the demographic data, the medical history of childbirth, the progress of childbirth, and, most importantly, the attitudes of the women and their environment toward childbirth. These were the main variables:
✓demographic data: level of education, status of relationship, age, place of residence;✓childbirth information: mode of delivery, indications for a cesarean section (if applicable), Apgar score for neonates, skin-to-skin contact, childbirth experiences, assessment of the decision about mode of delivery (if applicable), lactation experience;✓women’s perspectives: factors influencing the mode of delivery, methods of preparing for delivery.

All variables were included in the questionnaire, see Appendix A.

### 2.8. Statistical Analysis

STATISTICA version 13.3 statistical software (TIBCO Software Inc., Palo Alto, CA, USA) was used to analyze the collected data, by license of the Centre of Postgraduate Medical Education.

The following descriptive statistics were used to characterize the study group and variables according to the type of variable: count (n), relative frequency (%), mean (M), standard deviation (SD), maximum value (Max), and minimum value (Min).

Participants in the first stage of the study were divided into three groups according to their preference for the mode of delivery after a cesarean section:−Vaginal birth after cesarean section (VBAC) group—women who preferred vaginal childbirth;−Cesarean section preference (CSP) group—women who preferred a cesarean section due to a previous cesarean section;−No preference group—women who, at the time they completed the questionnaire, had no stated preference for the mode of birth in their current pregnancy.

Women participating in the second part of the study (after childbirth) were divided into four groups based on their answers to the question about the method of childbirth:−Emergency CS group—“I tried a vaginal delivery, but it ended with an emergency cesarean section”;−VBAC group—“I gave birth vaginally”;−Elective CS group—“I wanted to attempt a vaginal birth, but it was not possible due to medical reasons, so I had a planned cesarean section”;−Lack of consent for trial of labor after cesarean section (TOLAC) group—“I did not consent to a vaginal birth, instead I had a planned cesarean section.”

The Likert scale responses were ranked according to the following rule: definitely no (−2); rather no (−1); I have no opinion (0); rather yes (1), and definitely yes (2). The significance of differences was determined using the Kruskal–Wallis rank ANOVA statistic with independent samples or the Wilcoxon paired rank order test with dependent samples. Post hoc analysis was performed using the Dunn test. Comparative analysis of qualitative variables was performed using Pearson’s chi-squared test.

In the final stage of our statistical analysis, we utilized univariate and multivariate logistic regression models to discern the factors influencing the mode of childbirth after CS. The initial step involved conducting a univariate logistic regression analysis for each of the identified factors. The factors that demonstrated statistical significance in the univariate analysis were then incorporated into a multivariate logistic regression analysis. This allowed us to determine the independent effect of each factor on the preferred mode of childbirth while adjusting for potential confounders. Odds ratios (OR) and their corresponding 95% confidence intervals (CI) were calculated to quantify the strength of the associations.

For all analyses, a statistical significance level of 0.05 was assumed, meaning that the statistical null hypothesis was rejected for all results with *p* < 0.05.

## 3. Results

### Study Group Characteristics

The study included 288 pregnant women who had previously undergone childbirth by cesarean section. Sociodemographic data are presented in (Appendix A).

Among the women who preferred a vaginal birth, nearly 80% tried it and 49.78% had a natural childbirth. Among these women. 21.78% had a planned cesarean section for medical reasons. A cesarean section due to disagreement with a vaginal birth was performed in 52.78% of the women and due to medical reasons in 16.67% of the survey participants. More than 30% of the women who previously preferred CS tried vaginal childbirth, and 13.89% achieved VBAC. Forty percent of the women who declared no preference in the first stage of the survey tried vaginal delivery. One in four did not consent to TOLAC, and 36% had a cesarean section performed for medical reasons. The methods of delivery differed significantly according to the declared preference *p* < 0.001 (Figure 3).

No statistically significant differences were found in the baby’s birth weight according to the women’s preferences and the mode of delivery. Babies differed significantly in Apgar scale scores. The lowest scores (0–3 points) were most often received by neonates who were born by cesarean section, which was performed due to the woman’s choice (7.41%). In the elective cesarean section group, more neonates had Apgar scores between 8–10 points in comparison to the emergency cesarean section group (98.44% vs. 89.33% *p* = 0.017).

Statistically significant differences were also found for uninterrupted 2 h skin-to-skin contact. Babies born by vaginal delivery were more likely to have skin-to-skin contact with their mothers after birth. This contact was least frequent in the women who had an emergency cesarean section (*p* < 0.001) (Appendix A).

Significant differences were found in the women’s assessment of their decisions regarding the mode of delivery. More than 95% of the women who had a vaginal birth and almost 90% of the women who had an elective cesarean rated their decision well or very well. A quarter of the women who gave birth by elective cesarean section for medical reasons and almost as many women who attempted vaginal delivery but the birth ended in a cesarean section were unable to rate the decision. Nearly 40% of the women in each of these groups declared that the method of birth was not their decision (*p* < 0.001).

The experience of lactation also varied significantly depending on the mode of delivery. The women who had a vaginal birth had the best experience in this regard—more than 95% of the women rated it good or very good. The women who had an elective cesarean section—due to lack of consent to TOLAC (18.52%) and for medical reasons (17.19%)—rated this experience the worst (*p* < 0.001).

Birth preference toward a hypothetical subsequent pregnancy differed significantly according to the mode of the current labor. The women who had a natural childbirth in a subsequent pregnancy also preferred a vaginal birth for their next pregnancy (89.34%). This route was preferred by less than half of the women who had an intrapartum emergency cesarean section (46.67%) and a planned cesarean section for medical reasons (42.19%). Another elective cesarean section was most often chosen by the women who had a cesarean section due to lack of consent to TOLAC (59.26%). One-quarter of the women who had an emergency cesarean section and as many who had a planned cesarean section for medical reasons could not determine their preference for subsequent labor (*p* < 0.001) (Appendix A).

The women were also asked about the factors they would use in choosing a mode of delivery for their next pregnancy. The Kruskal–Wallis ANOVA test showed statistically significant differences for some of the factors depending on the mode of delivery. Post hoc tests revealed almost all of the differences were between the group of women who did not consent to TOLAC and all or some of the other groups. Although statistically significant differences were shown between all groups for the factor “convenience and predictability of CS,” the largest difference was also in this group. The experience of the previous birth significantly varied between the ECS and VBAC groups (Table 1 and Appendix A).

When asked what was most helpful in preparing for childbirth, the women most often answered that it was choosing a hospital where the staff supported their decision (63.19%). In second place was carrying a pregnancy with a supportive doctor (45.49%), and in third place was substantive preparation, i.e., reading up on the subject (38.54%). Slightly fewer women declared that enrollment in support groups was important (36.11%). This method of preparation was significantly more often declared by the women who tried vaginal childbirth. The ECS and lack of consent groups declared the importance of carrying a pregnancy with a supportive doctor significantly more often. The other modes of preparation did not differ significantly according to the mode of delivery (Appendix A).

The factors influencing the women’s decisions on the mode of delivery mostly underwent significant changes after labor. All factors except “better conditions for breastfeeding” scored higher after labor than before labor, indicating greater importance for the women. The largest differences (greater than 0.5) were observed for the following factors: the desire to minimize pain, the severity of the cesarean operation, and less blood loss. The next factors that became significantly more important to the women were: the convenience and predictability of the cesarean section, the impact of the birth mode on subsequent pregnancies and deliveries, and improved bonding with the partner (Table 2 and Appendix A).

In the following stage of the analysis, logistic regression was undertaken to identify the factors influencing the mode of childbirth. This analysis included questions related to the reasons that the women identified as guiding their preference for the method of delivery. The initial step involved a univariate analysis to examine the association between each of these reasons and the actual mode of childbirth (Appendix A). In the subsequent step, adjustments were made in the analysis for confounding factors such as age, education level, place of residence, and marital status.

In the adjusted univariate logistic regression analysis, the factors “convenience and predictability of ECS” (aOR 0.55, 95% CI 0.42–0.76, *p* = 0.001) and “ensuring better health of the mother” (aOR 1.42, 95% CI 1.10–1.88, *p* = 0.010) showed significant associations with the mode of childbirth. An increase in the factor “convenience and predictability of ECS” was associated with a 45% decrease in the probability of VBAC. On the other hand, the factor “ensuring better health of the mother” was associated with a 42% increase in the odds of VBAC.

The factor “severity of cesarean section” also demonstrated a significant association (aOR 1.21, 95% CI 1.01–1.45, *p* = 0.039), indicating a 21% increase in the odds of VBAC per unit increase in this factor. While the factor “ensuring better health for the child” showed an increase in the odds of VBAC (aOR 1.33, 95% CI 1.01–1.78), the *p*-value was slightly above the conventional threshold for statistical significance (*p* = 0.051). None of the other factors demonstrated significant associations with the actual mode of childbirth in this adjusted univariate logistic regression analysis (*p* > 0.05) (Table 3).

In the final stage of analysis, multivariate logistic regression was conducted, incorporating only those factors that were statistically significant in the univariate regression. Here, the factor “convenience and predictability of ECS” emerged as an independent factor significantly influencing natural birth after CS (OR 0.60, 95% CI 0.42–0.85, *p* = 0.004), suggesting a 40% decrease in the odds of VBAC with each unit increase in this factor. Meanwhile, the factors “severity of cesarean section” (OR 1.08, 95% CI 0.89–1.31, *p* = 0.425), “ensuring better health for the child” (OR 1.04, 95% CI 0.71–1.52, *p* = 0.832), and “ensuring better health of the mother” (OR 1.15, 95% CI 0.79–1.67, *p* = 0.479) did not demonstrate statistically significant associations with the mode of childbirth in this adjusted analysis (Table 4).

## 4. Discussion

Nearly 70% of the study’s participants tried a vaginal delivery after a previous CS, and more than 40% finally gave birth vaginally. This result was higher than in other studies, in which 63% and 40% of women tried a vaginal delivery and 40% and 30% achieved a natural childbirth, respectively [20,21]. These differences may be related to the methodology used in the study. This was not a cohort study, which would include a holistically defined group of women. The study included women interested in the topic of childbirth after cesarean section. According to the analysis presented in our previous study, those who preferred VBAC were more actively seeking information on the topic on the internet, so they may have been more likely to respond to the invitation to participate in the study than women who preferred birth by cesarean section [22].

At the same time, in the present study, only one in ten women declared that they had a cesarean section due to disagreement with vaginal birth. This represented about 12% of cesarean sections. This result was lower than that obtained in other studies conducted in Poland, in which the rate ranged from 17% to 34% [23,24,25,26]. The reason for such a low rate may have been due to the method of data collection described above, as well as the fact that a new organizational standard for perinatal care was issued and began to be implemented during the course of the study, according to which the possibility of a woman’s lack of consent was excluded [6]. The incidence of other indications for elective and emergency cesarean sections among the study participants was analogous to the results of other studies conducted in Poland [23,24,25,26].

The relationship between preference and the actual mode of delivery was slightly different than in previously conducted studies. In the present study, the percentage of women who preferred natural childbirth and eventually tried vaginal birth was 70%. In other studies, the preferences for natural birth were much lower at 56% and 40%. However, an opposite relationship was seen in the success rate of trial of labor, which in other studies was more than 70% while in the present study it was 60% [20,21]. We cannot explain the lower success rate based on the questionnaire. It may be due to several factors such as inadequate risk assessment for TOLAC and the inexperience of medical personnel in assisting TOLAC.

Of all the participants in the study who preferred natural childbirth after a cesarean section, as many as 40% eventually had birth by cesarean section. Some tried natural childbirth, which ended in an emergency cesarean section, and some had an elective CS due to maternal or child conditions. Research by McGrath et al. indicated that women in this situation were frustrated and disappointed that their bodies were unable to give birth naturally. The women’s perspectives shown in this study broke the stereotype that women who had birth by cesarean section were driven by ease and convenience. Rather, many times birth by this method was perceived by women to be a failure [27]. At the same time, the study by Kaimal et al. showed that women preferring VBAC were willing to try vaginal delivery even when there was more than 70% probability that it would end in a cesarean section. In contrast, women who preferred ECS were willing to try vaginal delivery if the risk of an emergency cesarean section was less than 35%. Once again, the trend was that women’s personal beliefs and attitudes were more important to them than objective medical conditions [20].

Both the women who gave birth vaginally and by cesarean section due to disagreement with TOLAC overwhelmingly rated their decisions as good or very good. About 40% of the women who had an emergency cesarean section and an equal number of the women who had an elective CS for medical reasons said that the mode of delivery was not their decision. It is interesting to point out that 40% of the women who tried vaginal childbirth but ended in an emergency cesarean section believed that trial of labor was a good or very good decision. These were different results from those presented in McGrath’s study stating that women who wanted to try vaginal childbirth but gave birth via cesarean section felt disappointment and even anger at their bodies and in their judgment at being incapable of achieving a natural birth [27]. Trying vaginal delivery can also carry psychological benefits. The women’s statements on the online forums described a sense of having done everything they could. Such action gave a sense of empowerment, even if the birth did not end as they preferred. It can also be assumed that such an attempt to give birth allowed them to feel the “real birth” and experience the pain, which included them in the circle of women who had actively given birth. However, this topic requires further research to understand the cause-and-effect relationships.

In our study, uninterrupted 2 h skin-to-skin contact with the mother was more common in babies born by natural childbirth and least common in those born by emergency cesarean section. These were the same correlations that have been shown in other studies involving all births, not just those by cesarean section [28,29]. Thus, it can be concluded that the problem of lack of recommended skin-to-skin contact or its implementation to a limited extent is not related to birth after a cesarean section but to the mode of delivery. Currently, efforts are being made around the world to ensure safe skin-to-skin contact for as many babies born by cesarean section as possible [30,31].

The same relationship occurred in the breastfeeding experience. A systematic review of the factors influencing the initiation and continuation of natural lactation based on 47 papers found that the mode of delivery was a crucial factor. Better lactation experience was had by women who achieved vaginal birth [32]. Thus, the issue of lactation, like skin-to-skin contact, in the case of labor after a cesarean section should be met with the same conditions as in other deliveries.

Analysis of the factors influencing the preferred mode of delivery was quite interesting. The women who disagreed with TOLAC were significantly less likely to indicate the importance of factors negatively influenced by CS, such as better conditions for breastfeeding or subsequent labor. These findings were consistent with those obtained by Bondar et al. [33]. However, an interesting field for further research is to find answers to questions such as: does the low importance of breastfeeding and the impact of a cesarean section on subsequent pregnancies affect the preference for mode of delivery? Or is the primary preference for a cesarean section and the declaration of low importance of the mentioned factors a form of self-justification for such a choice?

In our study, nearly 90% of the women who gave birth by vaginal delivery declared that if they had subsequent pregnancies they would also want to give birth by this method. Only about 30% of the women who gave birth by cesarean section wanted to do it again. It should be noted that almost half of the women wanted to have a vaginal birth after having had two cesarean sections. No studies have yet been conducted among women after two cesarean sections examining their attitudes toward subsequent births. Perhaps the well-established preference for natural childbirth is so strong that even a repeat cesarean section does not change it. These are different results from those among women who were in their first pregnancy, among whom a large proportion originally preferred natural childbirth and declared a change in preference to a second CS after giving birth by cesarean section [34].

The analysis of the women’s attitudes and opinions at two points in time—before and after childbirth—made it possible to observe the occurring changes. A big change could be observed in the modes of preparation declared before childbirth. The most useful, in the opinion of the women, was choosing a hospital that supported their decision, followed by carrying out the pregnancy with a supportive clinician. These two factors ranked outside the top three indications before childbirth. Substantive preparation, reading other women’s stories, and using support groups were no longer as important to women after giving birth as they were before. The benefit of support groups remained important for the group of women who had natural childbirth.

There were also significant changes in the importance of the individual factors declared as important for choosing the mode of delivery. All factors received significantly higher scores in the second stage of the study. Thus, it can be concluded that the experience of childbirth revised the women’s opinions on this issue. In particular, medical factors such as minimizing pain, the severity of the CS operation, and less blood loss became more important from a postpartum perspective.

The two above-mentioned findings are difficult to discuss within the existing literature, as to our best knowledge there has been no longitudinal quantitative study to date investigating the preferences and experiences of pregnant and postpartum women after a previous cesarean section.

The active participation and viewpoints of women regarding their mode of delivery are crucial components for shaping both medical management strategies and directions for future research. To develop a more holistic understanding of this issue, future investigations should consider launching cohort studies encompassing a diverse range of women, not merely those who have previously experienced a cesarean section (CS). This approach could provide a more in-depth understanding of the factors that influence preferences and outcomes concerning the mode of delivery. Additionally, future research should aim to untangle the cause-and-effect relationships that link women’s delivery preferences, actual delivery outcomes, and subsequent psychological well-being. This would require exploring elements such as personal beliefs, attitudes, and the role of medical conditions in decision-making processes. In particular, it would be insightful to assess the attitudes and preferences of women who have undergone two or more CS, given the rising prevalence of CS worldwide.

To aid the decision-making process, it is recommended that accessible and accurate information regarding childbirth options post-CS is provided, specifically for those women considering vaginal birth after cesarean (VBAC). Ensuring that women are well-informed could potentially result in more satisfying birth experiences. Additionally, psychological support and counseling should be offered, especially to those women who ended up having an emergency CS after expressing a preference for a vaginal birth following a CS. Addressing potential feelings of disappointment or failure could foster a more positive birthing experience. By adopting these recommendations and pursuing the suggested areas of research, healthcare professionals can better support women considering VBAC, improve the overall birthing experience, and promote evidence-based decision-making in clinical practice.

### Strengths and Limitations

A strength of the study was the use of a longitudinal cohort design. However, this study had some limitations. The first limitation was the lack of official statistics about the scale of the examined problem in Poland. The size of the population of pregnant women after cesarean section was estimated based on data on the number of pregnant women, the order of pregnancy, and the percentage of cesarean sections. Another limitation was the method of recruiting women for the study, which was conducted by sharing the questionnaire with a wide audience via the internet. Because the sample was not random, the entire population from the data presented may be affected by bias. Longitudinal studies further imply a limitation related to the possibility of participants dropping out of subsequent stages of the research. In the second stage of the analyzed research, only 40% of the women participating in the first stage took part, and the main reason was the lack of response to the twice-repeated request to complete the questionnaire. Therefore, although 733 women were included in the study, the final sample size of the longitudinal study was low. In addition, since the sample was not distributed randomly, inferences about the whole population based on the data presented could be affected by selection and recall biases.

The time frame for the second wave of data collection (6 weeks to 6 months) was fairly long, and the women’s opinions about their childbirth experience and future childbearing plans may have changed during this critical time. Therefore, the opinions of our study group could be heteronomous.

## 5. Conclusions

The mode of birth was not always the woman’s preference. Some of the women who preferred a vaginal delivery after a previous CS qualified for an elective cesarean section for medical reasons known before labor, while some had an emergency intrapartum cesarean section. Most of the women tried a vaginal birth, but ultimately one in four women gave birth by this method. Most of the women were satisfied with their choice of mode of delivery. Additionally, the women who tried a vaginal birth and had an emergency intrapartum cesarean section positively evaluated this decision.

A variety of changes were noticeable regarding the motivations, opinions, and attitudes among the women who gave birth after a cesarean section. A large proportion of the women who gave birth for a second time by cesarean section declared that they would like to have a natural birth in their next pregnancy. This is a field that warrants further exploration to better understand the mechanisms affecting women’s preferences, as it appears that these may be influenced by mostly non-medical factors.

## Figures and Tables

**Figure 1 healthcare-11-01806-f001:**
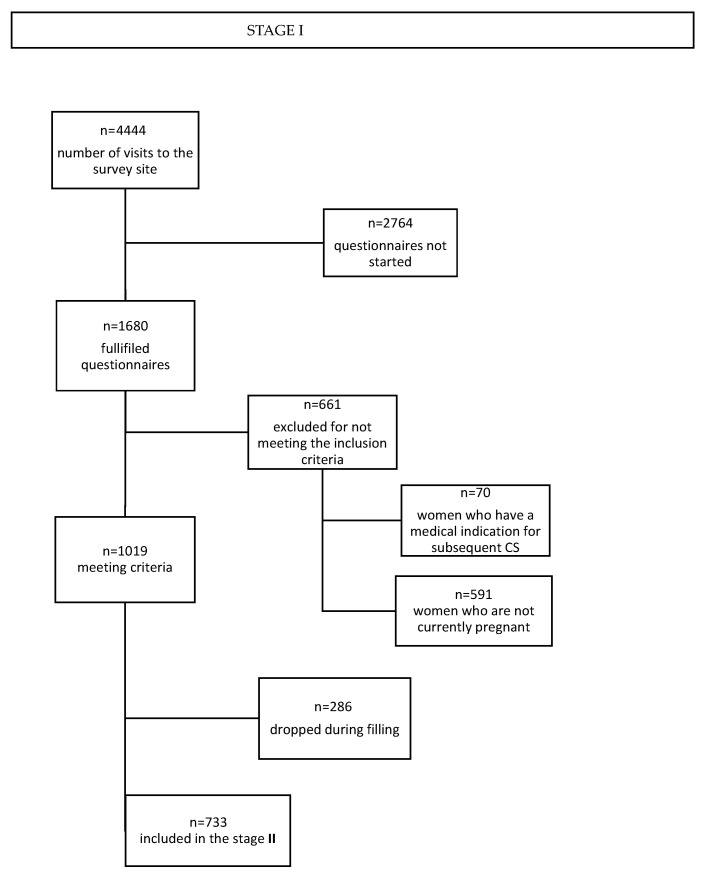
The scheme for recruiting participants for the study—stage I.

**Figure 2 healthcare-11-01806-f002:**
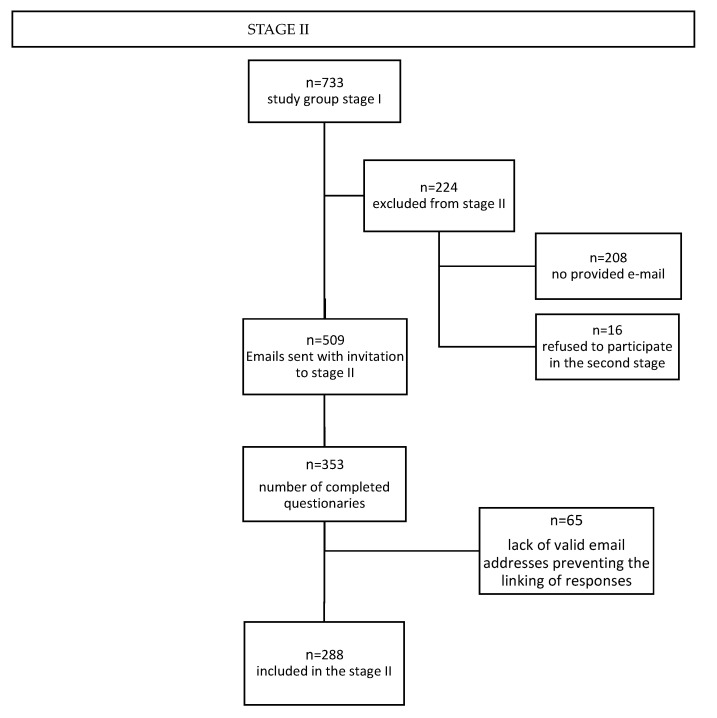
The scheme for recruiting participants for the study—stage II.

**Figure 3 healthcare-11-01806-f003:**
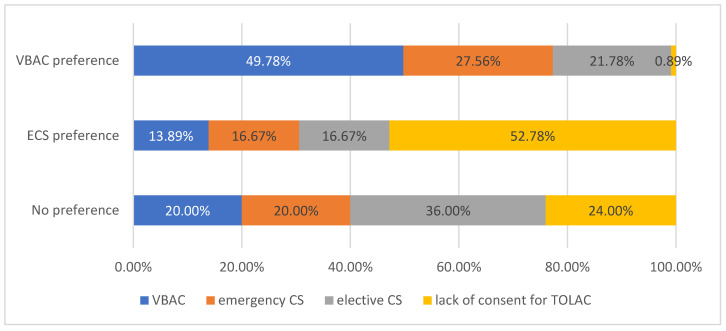
The actual mode of birth vs. the preferred mode of birth (*n* = 288) (*p* < 0.001). CS—cesarean section; VBAC—vaginal birth after cesarean section; TOLAC—trial of labor after cesarean section.

**Table 1 healthcare-11-01806-t001:** Factors in choosing the mode of delivery in subsequent pregnancies according to the mode of delivery after cesarean section.

	(1) Emergency CS *n* = 75		(2) VBAC *n* = 122		(3) Elective CS *n* = 64		(4) Lack of Consent for TOLAC *n* = 27		H	*p*	Post-Hoc
	M	SD	M	SD	M	SD	M	SD			
Minimizing pain	0.05	1.46	−0.12	1.46	−0.16	1.47	0.37	1.62	2.926	0.403	N/A
Convenience and predictability of ECS	−1.41	1.15	−1.47	1.11	−1.06	1.04	1.07	1.47	64.997	<0.001	ALL
Severity of cesarean section	0.97	1.35	1.18	1.23	0.80	1.45	−0.30	1.30	26.874	<0.001	1 vs. 4 2 vs. 4 3 vs. 4
Better bond between mother and child	1.19	1.15	0.97	1.41	0.91	1.43	0.19	1.44	10.820	0.013	1 vs. 4
Ensuring better health for the child	1.57	0.74	1.47	0.99	1.50	0.87	1.15	1.29	1.771	0.621	N/A
Ensuring better health of the mother	1.56	0.72	1.47	0.94	1.38	0.98	0.85	1.41	6.884	0.076	N/A
Ensuring skin-to-skin contact	1.63	0.83	1.62	0.94	1.53	0.89	0.96	1.34	11.487	0.009	-
Better conditions for breastfeeding	1.01	1.26	1.10	1.24	1.11	1.25	0.26	1.40	11.425	0.009	2 vs. 4 3 vs. 4
Influence of the mode of birth on subsequent pregnancies and deliveries	1.45	1.06	1.18	1.27	1.25	1.18	−0.07	1.57	24.613	<0.001	1 vs. 4 2 vs. 4 3 vs. 4
Faster recovery	1.55	0.83	1.49	0.98	1.47	0.94	0.37	1.52	22.976	<0.001	1 vs. 4 2 vs. 4 3 vs. 4
Less blood loss	0.83	1.27	0.55	1.32	0.56	1.22	0.00	1.44	7.659	0.054	N/A
Sense of fulfillment	1.29	1.12	1.20	1.30	0.89	1.46	−0.15	1.61	20.856	<0.001	1 vs. 4 2 vs. 4
Strengthening the sense of femininity	0.81	1.48	0.80	1.47	0.38	1.54	−0.26	1.63	12.923	0.005	1 vs. 4 2 vs. 4
Improving your relationship with your partner	−0.04	1.46	−0.02	1.43	−0.31	1.40	−0.26	1.63	2.460	0.483	N/A
Previous birth experiences	1.31	0.97	1.59	0.86	1.06	1.21	1.19	1.42	15.028	0.002	2 vs. 3
Previous postpartum experiences	0.80	1.25	1.01	1.28	0.83	1.33	0.78	1.58	2.636	0.451	N/A

CS—cesarean section; VBAC—vaginal birth after cesarean section; TOLAC—trial of labor after cesarean section; ECS—elective cesarean section; M—mean, SD—standard deviation, N/A—not applicable. Scale: definitely no (−2); rather no (−1); I have no opinion (0); rather yes (1), definitely yes (2).

**Table 2 healthcare-11-01806-t002:** Factors influencing the choice of mode of delivery—before and after labor.

	Before Labor		After Labor		z	*p*
	M	SD	M	SD		
Minimizing pain	−0.74	1.23	−0.02	1.47	5.790	<0.001
Convenience and predictability of ECS	−1.49	1.02	−1.12	1.35	3.301	0.001
Severity of cesarean section	0.40	1.44	0.90	1.38	4.946	<0.001
Better bond between mother and child	0.72	1.44	0.93	1.38	2.256	0.024
Ensuring better health for the child	1.34	0.96	1.47	0.94	2.071	0.038
Ensuring better health of the mother	1.24	1.01	1.41	0.97	2.790	0.005
Ensuring skin-to-skin contact	1.34	1.10	1.54	0.96	3.228	0.001
Better conditions for breastfeeding	1.02	1.24	1.00	1.28	0.024	0.981
Influence of the mode of birth on subsequent pregnancies and deliveries	0.84	1.47	1.15	1.30	3.716	<0.001
Faster recovery	1.29	1.14	1.40	1.05	1.320	0.187
Less blood loss	0.06	1.34	0.57	1.31	4.841	<0.001
Sense of fulfillment	0.89	1.46	1.02	1.39	2.070	0.038
Strengthening the sense of femininity	0.43	1.58	0.60	1.53	2.027	0.043
Improving your relationship with your partner	−0.49	1.31	−0.13	1.45	3.773	<0.001
Previous birth experiences	1.25	1.10	1.36	1.05	1.601	0.109
Previous postpartum experiences	0.76	1.37	0.88	1.31	1.285	0.199

ECS—elective cesarean section, M—mean, SD—standard deviation. Scale: definitely no (−2); rather no (−1); I have no opinion (0); rather yes (1), definitely yes (2).

**Table 3 healthcare-11-01806-t003:** Univariate logistic regression analysis for factors influencing the mode of childbirth after cesarean section (adjusted to: age, level of education, place of residence, education, marital status).

	aOR	95% CI		*p*
Minimizing pain	0.90	0.73	1.11	0.329
Convenience and predictability of ECS	0.55	0.42	0.76	**0.001**
Severity of cesarean section	1.21	1.01	1.45	**0.039**
Better bond between mother and child	0.95	0.80	1.12	0.515
Ensuring better health for the child	1.33	1.01	1.78	**0.051**
Ensuring better health of the mother	1.42	1.10	1.88	**0.010**
Ensuring skin-to-skin contact	1.13	0.89	1.44	0.324
Better conditions for breastfeeding	1.15	0.94	1.42	0.187
Influence of the mode of birth on subsequent pregnancies and deliveries	1.11	0.93	1.32	0.256
Faster recovery	1.20	0.95	1.53	0.126
Less blood loss	1.00	0.84	1.21	0.966
Sense of fulfillment	1.10	0.92	1.32	0.290
Strengthening the sense of femininity	1.10	0.94	1.29	0.244
Improving your relationship with your partner	1.06	0.87	1.28	0.571
Previous birth experiences	1.04	0.83	1.31	0.752
Previous postpartum experiences	0.98	0.83	1.17	0.850

aOR—adjusted Odds Ratio, CI—Confidence Interval.

**Table 4 healthcare-11-01806-t004:** Multivariate logistic regression analysis of independent factors influencing mode of childbirth after cesarean section.

	OR	95% CI		*p*
Convenience and predictability of ECS	0.60	0.42	0.85	0.004
Severity of cesarean section	1.08	0.89	1.31	0.425
Ensuring better health for the child	1.04	0.71	1.52	0.832
Ensuring better health of the mother	1.15	0.79	1.67	0.479

OR—Odds Ratio, CI—Confidence Interval.

## Data Availability

The data presented in this study are available upon request from the corresponding author.

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
