# Peer review of "The Experience of Women Giving Birth after Cesarean Section—A Longitudinal Observational Study"

_healthcare, 2023, doi:10.3390/healthcare11121806_

Round 1

Reviewer 1 Report

The authors studied women’s feelings, attitude, satisfaction, and so on which could vary according to situation around delivery. This is a very important issue for all women and perinatal medical staffs, and of clinical value. Their study would be interesting to possible readers of the journal healthcare but requires some amendment before publication.

*Methods

l.125 The contents of the first questionnaire is required as a supplementary material.

l.130 The contents of the second questionnaire is required as a supplementary material.

l.137 The names and URLs of social media and parenting portals used in this study is required as a supplementary material.

l.173 It is better to spell out “chi2” as “chi-square”.

*Results

Table 1 Subheadings like “level of education” should be left-justified to be more reader-friendly.

Table 1 The second decimal place should be rounded. (The same for throughout the entire manuscript.)

Figure 1 “rejected” seems to have to be replaced by “excluded”.

l.212 “The least number…” is not proper, because the actually least is 25 observed in the lack of consent for TOLAC group. Please correct it.

l.212 “the highest scores” should be inserted after “received”, or brackets of “(8-10 points)” removed.

Table 4 A partial application of bold fonts is inappropriate, potentially leading readers to biased understanding of the table.

Table 5 A partial application of bold fonts is inappropriate, potentially leading readers to biased understanding of the table.

*Discussion

l.305 “eventually tried childbirth” should be corrected as “eventually tried vaginal birth” or “eventually tried it”, because cesarean delivery is also childbirth.

l.306 Please clarify what the figures of 56% and 40% respectively stands for.

l.308 “success rate of the trial of vaginal labor” should be corrected as “success rate of vaginal delivery”.

l.312 “surgical labor” is replaced with a proper expression (CS).

*Others

- Usage of the word “labor” seems inappropriate and so confusing. Labor often means pain of uterine contraction during delivery. Labor can be used to mean vaginal delivery but is not usually mean Cesarean delivery. Thus, expressions such as “surgical labor”, “route of labor” and even “mode of labor” are not appropriate. Most of “labor” can be safely replaced with “delivery”.

- Usage of the expression “elective CS” is not appropriate and confusing. Elective CS means planned CS with any reason. When a woman with a previous CS would not like TOLAC and plan CS, that is an elective CS. So, the expression (the name of a subgroup) should be changed, for example, to “elective CS with medical indication”.

- The expressions “lack of consent” and “disagreement” imply the authors’ preference for vaginal delivery over CS, which means they are biased expressions. CS should not be a negative activity from a medical perspective, even if women or perinatal medical staffs feel so. Please revise them.

- Some typos and grammatical or wording errors are seen. Both “labor” and “labour” are used. Please make a thorough inspection of the entire manuscript. Proofreading by native English users or some paid service is advised.

Author Response

Dear Reviewer,

Thank you very much for your thoughtful comments and valuable insights. We appreciate the time you have taken to review our manuscript and agree that the issues raised are critical for all women and perinatal medical staff. We also acknowledge the potential clinical value of our study.

Following your suggestions, we have revised our manuscript to ensure a more cohesive and comprehensive presentation. We agree with your assessment that our study will be of interest to the readers of the Healthcare journal, and we have made amendments accordingly to improve the readability and overall quality of the work.

In the following, we address each of your points in detail:

*Methods

l.125 The contents of the first questionnaire is required as a supplementary material.

added Appendix A

l.130 The contents of the second questionnaire is required as a supplementary material.

added Appendix B

l.137 The names and URLs of social media and parenting portals used in this study is required as a supplementary material.

added Appendix C

Information about appendices has been added to the manuscript.

l.173 It is better to spell out “chi2” as “chi-square” – corrected

*Results

Table 1 Subheadings like “level of education” should be left-justified to be more reader-friendly. - corrected

Table 1 The second decimal place should be rounded. (The same for throughout the entire manuscript.)

Three decimal places have only test results and p-values, as this is how they are commonly accepted and used. All other values are in two decimal places.

Figure 1 “rejected” seems to have to be replaced by “excluded”. - corrected

l.212 “The least number…” is not proper, because the actually least is 25 observed in the lack of consent for TOLAC group. Please correct it. - rephrased

l.212 “the highest scores” should be inserted after “received”, or brackets of “(8-10 points)” - rephrased

Table 4 A partial application of bold fonts is inappropriate, potentially leading readers to biased understanding of the table. - corrected

Table 5 A partial application of bold fonts is inappropriate, potentially leading readers to biased understanding of the table. - corrected

*Discussion

l.305 “eventually tried childbirth” should be corrected as “eventually tried vaginal birth” or “eventually tried it”, because cesarean delivery is also childbirth - corrected

l.306 Please clarify what the figures of 56% and 40% respectively stands for - corrected

l.308 “success rate of the trial of vaginal labor” should be corrected as “success rate of vaginal delivery” - corrected

l.312 “surgical labor” is replaced with a proper expression (CS) - corrected

*Others

- Usage of the word “labor” seems inappropriate and so confusing. Labor often means pain of uterine contraction during delivery. Labor can be used to mean vaginal delivery but is not usually mean Cesarean delivery. Thus, expressions such as “surgical labor”, “route of labor” and even “mode of labor” are not appropriate. Most of “labor” can be safely replaced with “delivery”. – corrected as suggested throughout the manuscript

- Usage of the expression “elective CS” is not appropriate and confusing. Elective CS means planned CS with any reason. When a woman with a previous CS would not like TOLAC and plan CS, that is an elective CS. So, the expression (the name of a subgroup) should be changed, for example, to “elective CS with medical indication”. – we have changed the name of group women who prefer cesarean section, due to a previous cesarean section to CS preference.

- The expressions “lack of consent” and “disagreement” imply the authors’ preference for vaginal delivery over CS, which means they are biased expressions. CS should not be a negative activity from a medical perspective, even if women or perinatal medical staffs feel so. Please revise them. We agree that this can be perceived as bias but as stated in the methods section: “The study was conducted in Poland while The Polish Society of Gynecologists and Obstetricians' recommendations allowing women to co-decision about the mode of labour after cesarean section were in effect. In the case of a physician's recommendation to try vaginal labour, the woman had the option not to consent to it. As a result, an elective cesarean section was performed.” As a result these cesarean section were performed for indictaions described as “Lack of consent for TOLAC”. In reality these were women that were offered TOLAC because they had no contraindications for trial of labour but did not consent to it i.e. did not attempt TOLAC. We can change the group name to Did not attempt TOLAC but we thing that the present name of the group is more in unison with the Polish recommendations.

- Some typos and grammatical or wording errors are seen. Both “labor” and “labour” are used. Please make a thorough inspection of the entire manuscript. Proofreading by native English users or some paid service is advised. – revised through the manuscript

We hope these revisions and responses address your concerns and look forward to your continued feedback. Thank you once again for your constructive review.

Sincerely,

Dorota Sys

Reviewer 2 Report

"The experience of women giving birth after cesarean section – a longitudinal observational study".  Introduction and discussion was written in detail. Some clarifications needed:

1. Line 102 - "Exclusion criterium was an indication for cesarean section in the current pregnancy". On what basis you decide this? Do you have any questions related to this?

2. Avoid unnecessary space in between numbers, instead insert a comma. Also don't add comma in place of full stop. For instance: lines 115, 116, 206, numbers in figure 1.

3. Line 122 - The questionnaires were created by an interdisciplinary team of experts that included midwives, obstetrician-gynecologists, and a sociologist. Did you prepare this questionnaire by literature review? Was this questionnaire validated? What is the validity score?

4. Line 155 - Elective cesarean section (ECS) group - women who prefer cesarean section, due to a previous cesarean section. In line 164, you mentioned 'elective CS group – “I wanted to attempt a vaginal birth, but it was not possible due to medical reasons, so I had a planned cesarean section”'. Both statements are contradictory. More over in one place you gave an abbreviation of ECS and in other place, as elective CS for a same term. When they had CS due to medical reasons, it was an exclusion criteria of our study - Justify.

5. In figure 1, you excluded 70 women who have a medical indication for subsequent CC. What is CC? Again this statement is related to my 1st and 4th question.

6. In table 1, you presented sociodemographic data. In what way are these data related to your study objective?

7. In figure 3, elective CS would have been excluded, as per your exclusion criteria right?

8. What is the association of table 2 with your study? Moreover, for a question, "Did you have skin-to-skin contact with your baby after birth?", the options were 'No', 'Yes, but it was less than 2 hours' and 'yes'. 'Yes' indicates more than 2 hours?

Author Response

Dear Reviewer,

Thank you very much for your thoughtful comments and valuable insights. We appreciate the time you have taken to review our manuscript and agree that the issues raised are critical for all women and perinatal medical staff. We also acknowledge the potential clinical value of our study.

Following your suggestions, we have revised our manuscript to ensure a more cohesive and comprehensive presentation. We agree with your assessment that our study will be of interest to the readers of the Healthcare journal, and we have made amendments accordingly to improve the readability and overall quality of the work.

In the following, we address each of your points in detail:

"The experience of women giving birth after cesarean section – a longitudinal observational study".  Introduction and discussion was written in detail. Some clarifications needed:

  1. Line 102 - "Exclusion criterium was an indication for cesarean section in the current pregnancy". On what basis you decide this? Do you have any questions related to this? - yes, questionnaire part 1 (Appendix A) – Question W2.
  2. Avoid unnecessary space in between numbers, instead insert a comma. Also don't add comma in place of full stop. For instance: lines 115, 116, 206, numbers in figure 1. - corrected
  3. Line 122 - The questionnaires were created by an interdisciplinary team of experts that included midwives, obstetrician-gynecologists, and a sociologist. Did you prepare this questionnaire by literature review? - yes Was this questionnaire validated? – yes. Validation of the questionnaire consisted of 3 stages: the first was a pilot study, the second was an evaluation by the researchers, the next stage was to implement corrections, and the final stage was to calculate validity using Cronbach's alpha..

 What is the validity score? - Alpha-Crombach=0,821

  1. Line 155 - Elective cesarean section (ECS) group - women who prefer cesarean section, due to a previous cesarean section. In line 164, you mentioned 'elective CS group – “I wanted to attempt a vaginal birth, but it was not possible due to medical reasons, so I had a planned cesarean section”'. Both statements are contradictory. More over in one place you gave an abbreviation of ECS and in other place, as elective CS for a same term. When they had CS due to medical reasons, it was an exclusion criteria of our study - Justify. we have changed the name of group women who prefer cesarean section, due to a previous cesarean section to CS preference. Elective cesarean section – women who had CS for medical indications
  2. In figure 1, you excluded 70 women who have a medical indication for subsequent CC. What is CC? Again this statement is related to my 1st and 4th question. - CS – typo mistake
  3. In table 1, you presented sociodemographic data. In what way are these data related to your study objective? These data have been shown to be significant factors in previous studies analyzing women's preferences for mode of delivery.
  4. In figure 3, elective CS would have been excluded, as per your exclusion criteria right?

This is a longitudinal study. Women filled the first questionnaire during pregnancy and the second after delivery. After the second questionnarie a group of women who planned vaginal delivery ended up having medical indications for cesarean delivery at the end of pregnancy. This resulted in an additional group of women.

  1. What is the association of table 2 with your study? Moreover, for a question, "Did you have skin-to-skin contact with your baby after birth?", the options were 'No', 'Yes, but it was less than 2 hours' and 'yes'. 'Yes' indicates more than 2 hours? 2 hours is the WHO recommendation. Apgar score is indicative of neonatal well-being.

We hope these revisions and responses address your concerns and look forward to your continued feedback. Thank you once again for your constructive review.

Sincerely,

Dorota Sys

Reviewer 3 Report

The authors examine an important topic that could have potentially significant clinical and public health implications for maternal health. However, the Methods (especially the analysis) and Results sections require substantial revision, and the Introduction and Discussion sections could benefit from additional information. Please see specific feedback below:

Abstract: In line 23, “women’s” has an extra “s”.

Introduction:

-       As the premise of the paper is the importance of women being able to have a VBAC (vaginal birth after c-section), information regarding the benefits of a vaginal delivery over a c-section should be included to further justify the need for this paper.

-       It is unclear what a cross-sectional longitudinal study is—these are two different study designs. Unless you can provide an explanation/definition for this study design in the paper, I would recommend using the term longitudinal cohort study throughout the paper.

-       The authors should highlight that a strength of the study is the use of a longitudinal cohort design.

Methods:

-       Importantly, a Measures section is missing. The authors need to outline and define all of the constructs that will be explored in the study as well as why they choose to operationalize constructs using the variables they selected. Currently, the Results section is challenging to get through because there is no prior discussion of the variables.

-       The time frame for the second wave of data collection (6 weeks to 6 months) is fairly long, and women’s opinions about their childbirth experience and future childbearing plans may change during this critical time. The authors should either control for time or at the very least, list this as a limitation of a study.

-       In line 101, what does status after previous c-section mean?

-       It is stated that 375 women should be in the sample based on power calculations, but only 288 women were included in the final analysis. The low sample size should be included as a limitation.

-       The Bias section information should be moved to the Limitations section in the Discussion.

-       The authors only conduct descriptive and bivariate analysis, which does not adjust for potential confounders such as maternal age, health conditions, and sociodemographic characteristics. If multivariate analysis cannot be conducted, this critical limitation needs to be included in the Limitations section.

-       In Table 1, please include category headings for each type of sociodemographic characteristic.

-       In the tables that contain means, it is unclear why there are negative means. Please clarify or revise.

-       In Table 5, different fonts are used for the information in the table—please use one consistent font.

-       In general, too many tables /data are presented, and it is challenging to get through. Perhaps highlight the most important information and move some of the tables to appendices.

Discussion:

-       The current information in this section could be more concise overall, but the authors do need to add a section regarding the clinical and public health implications of their findings as well as future research directions for this work.

-       The Limitations section needs to be substantially expanded, as previously described.

Author Response

Dear Reviewer,

Thank you very much for your thoughtful comments and valuable insights. We appreciate the time you have taken to review our manuscript and agree that the issues raised are critical for all women and perinatal medical staff. We also acknowledge the potential clinical value of our study.

Following your suggestions, we have revised our manuscript to ensure a more cohesive and comprehensive presentation. We agree with your assessment that our study will be of interest to the readers of the Healthcare journal, and we have made amendments accordingly to improve the readability and overall quality of the work.

We address each of your points in detail. Please see the attachment.

We hope these revisions and responses address your concerns and look forward to your continued feedback. Thank you once again for your constructive review.

Sincerely,

Dorota Sys

Reviewer 4 Report

Thank you for the opportunity to review this important work.

Abstract = assume this is referring to women who are having a subsequent child after a previous cesarean birth. This needs to be made clearer and would then help set the scene better.

Line 22 supported their decision for what - vaginal or cesarean birth

Line 25 ? vaginal birth not labour or is it vaginal birth who labour - not clear

Line 27 is this referring to women who have a VBAC and their preference for vaginal birth for the subsequent child.

Make last sentence a recommendation - so recommend that hospitals support  - sentence could be clearer

Like the use of the term natural childbirth

Line 39 change cooperate to maybe collaborate (or something else) - a more inclusive term

Line 40 - each birth method

Line 48 - not clear what 'condition' refers to

Line 53 ? and the wishes of the women

Line 56 wonder whether why this is an issue should be outlined here

Line 61 this statement refers to whatever the birth outcome

Line 79 aim needs to be clear and consider the point discussed in the abstract

Design how was the questionnaire developed, tested, validity and so forth (I see this is later but wondered if it should be here to be more logical) Assume ethics approval occurred.

Line 94 ? more about mode of childbirth or are you referring to actually having labour before c section

Line 96 ? vaginal birth

Line 101 not sure what 'status after previous c section' means

How were the women recruited as in how did you know these women were eligible to recruit - birth register, going through medical records or how. 

Line 107 was the second stage of the study the 6 weeks and 6 month or what was it

Line 116 this is a staggering c section rate and should be part of the argument presented earlier and a justification for the research

Line 124 need a bit more about pilot

Line 129 assume the second questionnaire is done at 6 weeks or is this another one to the one done after labour

Not clear what matrix questions are

Recruitment could be earlier to make more logical

Line 152 ? mode of birth not labour

Line 192 'finished labour' does this mean they had a vaginal birth - inconsistent to sentence in line197

Lie 193 ? occurred in 21>78% of these women

Line 194 proportions of what - to be clear

Line 195 sentence is not clear

Line 199 this is not clear - not sure if this is because labour and birth are used interchangeably

Line 201 - should this be birth not labour

Line 208 ? birth not labour

Line 209 babies or infant or neonates rather than children

Line 210 - could make this elective c section rather then what is written. Then confused with the next sentence and what the difference between these two groups are. This needs clarifying what the difference is.

Line 218 ? birth rather than labour

Line 222 vaginal birth rather then deliveries to be consistent

Line 228 is this labour or birth

Line 231 - disagreement with vaginal birth - not clear what this means, line 238 as well

Line 234 ? birth not labour

Line 235 'prefer this mode of birth for their next pregnancy'

Line 240 planned or elective - again consistency

Line 241 ? birth, same line 246, 247

Line 249, 250 now introduced new terms - be consistent

Line 271 I assume this is their preference for subsequent pregnancy childbirth

Line 283 ? birth or could put labour/birth

Line 284 gave birth vaginally or achieved a vaginal birth or natural birth

Line 285 - achieved a vaginal birth

Line 294 clarify disagreements

Line 303 labour or birth - same throughout this section

Line 308 sentence not clear

Line 313 need reference number

Line 323 not clear what disagreement with TOLAC means

Line 339 issue is more a staffing issue then anything

Line 362 birth

also an issue of at least having a labour no matter what the birth option is

line 407 ? birth not labour

This is very interesting research so the researcher should be congratulated for undertaking this work as it is a significant issues.

Line 236 intrapartum c section is this an emergency c section

Important to be consistent with language to make this clear

Author Response

Dear Reviewer,

Thank you very much for your thoughtful comments and valuable insights. We appreciate the time you have taken to review our manuscript and agree that the issues raised are critical for all women and perinatal medical staff. We also acknowledge the potential clinical value of our study.

Following your suggestions, we have revised our manuscript to ensure a more cohesive and comprehensive presentation. We agree with your assessment that our study will be of interest to the readers of the Healthcare journal, and we have made amendments accordingly to improve the readability and overall quality of the work.

In the following, we address each of your points in detail:

Abstract = assume this is referring to women who are having a subsequent child after a previous cesarean birth. This needs to be made clearer and would then help set the scene better.- corrected as suggested.

Line 22 supported their decision for what - vaginal or cesarean birth - regardless of what the decision was – added this information

Line 25 ? vaginal birth not labour or is it vaginal birth who labour - not clear - correcxtewd

Line 27 is this referring to women who have a VBAC and their preference for vaginal birth for the subsequent child. – yes.

Make last sentence a recommendation - so recommend that hospitals support  - sentence could be clearer –

We have added this sentence: Hospitals should support women's birth preferences after a cesarean section (if medically appriopriate), providing comprehensive counseling, resources, and emotional support to ensure informed decisions and positive birth experiences

Like the use of the term natural childbirth – rephrased as suggested

Line 39 change cooperate to maybe collaborate (or something else) - a more inclusive term – changed as suggested

Line 40 - each birth method – changed as suggested

Line 48 - not clear what 'condition' refers to – rephrased to clearly

Line 53 ? and the wishes of the women – yes, added

Line 56 wonder whether why this is an issue should be outlined here - We agree that this is very interesting and it would be useful to conduct an in-depth analysis of this situation - however, I am afraid that discussion of this topic here would be out of scope

Line 61 this statement refers to whatever the birth outcome - yes

Line 79 aim needs to be clear and consider the point discussed in the abstract - rephrased as suggested

Design how was the questionnaire developed, tested, validity and so forth (I see this is later but wondered if it should be here to be more logical) Assume ethics approval occurred. - We have used two standardized study reporting guides in our manuscript _ STROBE and CHERRIES - the content layout is in agreement with the requirements of these guides. If you agree, we would stay with this content layout

Line 94 ? more about mode of childbirth or are you referring to actually having labour before c section

Line 96 ? vaginal birth - changed as suggested

Line 101 not sure what 'status after previous c section' means - corrected

How were the women recruited as in how did you know these women were eligible to recruit - birth register, going through medical records or how. – this is described later – in the requirement subsection: Recruitment

The questionnaires were entered into an online platform (www.interaktywnie.com) that allows questionnaires to be shared online. Once the questionnaires were entered into the system, a link to each of the two questionnaires was generated allowing access to the survey.

Recruitment of pregnant women for the study was carried out through social media and parenting portals. A link with a description of the study was posted in open and closed groups and forums dedicated to pregnant and postpartum women. (see Appendix C) The scheme for recruiting participants for the study is shown in flow charts. (Figures 1 and 2)

Line 107 was the second stage of the study the 6 weeks and 6 month or what was it -between 6 weeks to 6 month after labor

Line 116 this is a staggering c section rate and should be part of the argument presented earlier and a justification for the research – yes we agree, this is like an epidemic of cesarean section, so this is one of the reasons we conducted this research.

Line 124 need a bit more about pilot

We have added information: Several questionnaires were distributed to pregnant women asking them to fill out the questionnaires and provide their comments, concerns, or difficulties in filling them out, if any. The comments were then analyzed by the research team and those that were applicable were incorporated.

Line 129 assume the second questionnaire is done at 6 weeks or is this another one to the one done after labour - -between 6 weeks to 6 month after labor

Not clear what matrix questions are - A Matrix question is a group of multiple-choice questions displayed in a grid of rows and columns. The rows present the questions to the respondents, and the columns offer a set of predefined answer choices that apply to each question in the row. Very often the answer choices are on a scale.

Recruitment could be earlier to make more logical – as we informed you earlier we are following  STROBE and CHERRIES guidance

Line 152 ? mode of birth not labour - corrected

Line 192 'finished labour' does this mean they had a vaginal birth - inconsistent to sentence in line197  - corrected

Lie 193 ? occurred in 21>78% of these women - -corrected

Line 194 proportions of what - to be clear - corrected

Line 195 sentence is not clear - corrected

Line 199 this is not clear - not sure if this is because labour and birth are used interchangeably – Rephrased: Forty percent of women who declared no preference in the first stage of the survey tried vaginal delivery

Line 201 - should this be birth not labour - corrected

Line 208 ? birth not labour- corrected

Line 209 babies or infant or neonates rather than children - corrected

Line 210 - could make this elective c section rather then what is written. Then confused with the next sentence and what the difference between these two groups are. This needs clarifying what the difference is.- rephrased to clarify

Line 218 ? birth rather than labour -  corrected

Line 222 vaginal birth rather then deliveries to be consistent birth - corrected

Line 228 is this labour or birth - corrected

Line 231 - disagreement with vaginal birth - not clear what this means, line 238 as well – lack of consent of TOLAC - corrected

Line 234 ? birth not labour - corrected

Line 235 'prefer this mode of birth for their next pregnancy' - corrected

Line 240 planned or elective - again consistency – emergency CS is intrapartum CS as described in the methods section

Line 241 ? birth, same line 246, 247 - corrected

Line 249, 250 now introduced new terms - be consistent - corrected

Line 271 I assume this is their preference for subsequent pregnancy childbirth – yes, this is.

Line 283 ? birth or could put labour/birth - corrected

Line 284 gave birth vaginally or achieved a vaginal birth or natural birth - corrected

Line 285 - achieved a vaginal birth - corrected

Line 294 clarify disagreements – as described: lack of consent of TOLAC (it can be according to Polish recommendation)

Line 303 labour or birth - same throughout this section – corrected as suggested

Line 308 sentence not clear – rephrased to:Based on the questionnaire we cannot explain the lower success rate. It may be due to several factors such as inadequate risk assessment for TOLAC, the inexperience of medical personnel in assisting TOLAC”

Line 313 need reference number – this is 23 number of reference – after 3 sentences

Line 323 not clear what disagreement with TOLAC means – it is described in the methodology section: – “I did not consent to a vaginal birth, I had a planned cesarean section”.

Line 339 issue is more a staffing issue then anything - it may be that way. unfortunately, we can only assume, this requires additional in-depth research

Line 362 birth - corrected

also an issue of at least having a labour no matter what the birth option is – yes, agree

line 407 ? birth not labour  corrected

This is very interesting research so the researcher should be congratulated for undertaking this work as it is a significant issues.

Thank you so much

Line 236 intrapartum c section is this an emergency c section – added emergency

Important to be consistent with language to make this clear- The whole manuscript has been corrected.

We hope these revisions and responses address your concerns and look forward to your continued feedback. Thank you once again for your constructive review.

Sincerely,

Round 2

Reviewer 2 Report

Dear Author

I didn't get satisfactory answer for some of my questions.

- "Exclusion criterium was an indication for cesarean section in the current pregnancy". On what basis you decide this indication? The question W2 in Appendix 1 is "Which mode of delivery do you prefer when you are currently pregnant?". This question is about the women's preference. My question is how do you exclude based on the indication for CS in current pregnancy. Give examples for indication.

- In table 2, for a question, "Did you have skin-to-skin contact with your baby after birth?", the options were 'No', 'Yes, but it was less than 2 hours' and 'yes'.

What does 'Yes' indicate? (After how long - more than 2 hours? Mention it)

Since WHO recommendation is less than 2 hours, you should have categorized into "No", "Yes", "yes, but it was more than 2 hours".

Author Response

Dear Reviewer,

Thank you for your opinion and additional questions. Please find our response point by point below.

"Exclusion criterium was an indication for cesarean section in the current pregnancy". On what basis you decide this indication? The question W2 in Appendix 1 is "Which mode of delivery do you prefer when you are currently pregnant?". This question is about the women's preference. My question is how do you exclude based on the indication for CS in current pregnancy. Give examples for indication.

In the APPENDIX A is the Question:

W2. [Z, max. 1]*

Which mode of delivery do you prefer when you are currently pregnant?

  1. vaginal birth
  2. caesarean section
  3. I have to have a c-section for medical reasons
  4. I have not yet thought about my preferred method of delivery in my current pregnancy

If a woman selected the answer “C. I have to have a c-section for medical reasons” the questionnaire was closed with the information that she don’t meet inclusion criteria.

There are examples of medical reasons:

abnormal position of the fetus (e.g. pelvic, oblique)

suspicion of macrosomia, feto-pelvic disproportion

maternal diseases related to pregnancy (e.g. cholestasis, diabetes, hypertension)

cesarean scar dehiscence after a previous cesarean section

placental abnormalities (e.g., placenta previa, abnormal invasion)

non-obstetric indications (e.g. cardiological, psychiatric, ophthalmological)

fetal indications (e.g. heart defects contraindicated in vaginal birth)

- In table 2, for a question, "Did you have skin-to-skin contact with your baby after birth?", the options were 'No', 'Yes, but it was less than 2 hours' and 'yes'.

What does 'Yes' indicate? (After how long - more than 2 hours? Mention it)

Since WHO recommendation is less than 2 hours, you should have categorized into "No", "Yes", "yes, but it was more than 2 hours".

WHO recommendation is at least 2 hour, therefore we think that our answers are suitable.

Best regards

Dorota Sys PhD

Reviewer 3 Report

Thank you for making these revisions--the paper is much improved. However, I have some additional feedback:

 - I am unable to see the appedices. Either way, I think that it would still be helpful to include a Measures section in the Methods - at the very list, please list the main variables and refer to the correct Appendix for further information. 

- Importantly, please note that Cronbach's alpha measure reliability, not validity. Along these lines, it ideally assesses the internal consistency of a scale/measure, not an entire questionnaire. If you are assessing validity, please discuss aspects of internal and external validity. 

- Please proofread lines 527-529--it appears to be two separate sentences. Also, the word "heteronomous" is confusing--please expand on this. 

- The low sample size is not included in the Limitations section.

- Again, none of the appendices are included in the manuscript. 

Author Response

Dear Editor,

Thank you for the revision of our manuscript and your opinion. We corrected the manuscript as suggested.  Please find point-by-point answers below:

- I am unable to see the appedices. Either way, I think that it would still be helpful to include a Measures section in the Methods - at the very list, please list the main variables and refer to the correct Appendix for further information. 

We have added information as sugggested:

Measures

The study measured a range of variables relating to the demographic data, the medical history of childbirth, the progress of childbirth, and, most importantly, the attitudes of women and their environment toward childbirth. There are main variables:

         demographic data: level of education, status of relationship, age, place of resi-dence;

         childbirth information: mode of delivery, indications for a cesarean section (if applicable), Apgar score for neonates, skin-to-skin contact, childbirth experi-ences: assess the decision about mode of delivery (if applicable), lactation ex-perience;

         women’s perspective: factors influencing of the mode of delivery, methods of preparing for delivery.

All varaibles are included in the questionnaires, see Appendix A and B.

- Importantly, please note that Cronbach's alpha measure reliability, not validity. Along these lines, it ideally assesses the internal consistency of a scale/measure, not an entire questionnaire. If you are assessing validity, please discuss aspects of internal and external validity. 

Yes, we agree with you. We rephrased this part of methods section:

Validation of the questionnaire consisted of two stages: the first was a pilot study, the second was an evaluation by the researchers, the next stage was to implement corrections. In the pilot study, several questionnaires were distributed to pregnant women asking them to fill out the questionnaires and provide their comments, concerns, or difficulties in filling them out, if any. The comments were then analyzed by the research team and those that were applicable were incorporated. In addition, we conducted a reliability analysis for the scale we used to find factors influencing mode of delivery.  Cronbach's alpha for this scale was 0.821.

- Please proofread lines 527-529--it appears to be two separate sentences. Also, the word "heteronomous" is confusing--please expand on this. 

We have rephrased  the limitation section:

A strength of the study is the use of a longitudinal cohort design. This study has some litmitation. The first limitation is the lack of official statistics about the scale of the examined problem in Poland. The size of the population of pregnant women after the cesarean section was estimated based on data on the number of pregnant women, the order of pregnancy, and the percentage of cesarean sections. Another limitation was the method of recruiting women for the study, which was done by sharing the questionnaire with a wide audience via the Internet. Because the sample was not random, inferring the entire population from the data presented may be affected by bias. Longitudinal studies further imply a limitation related to the possibility of participants dropping out of subsequent stages of the research. In the second stage of the analyzed research, 40% of the women participating in the first stage took part, and the main reason was the lack of response to the request to complete the questionnaire repeated twice. Therefore, although 733 women were included in the study, the final sample size of the longitudinal study is low. In addition, since the sample was not distributed ran-domly, inferences about the whole population based on the data presented could be affected by selection bias and recall bias.

  The time frame for the second wave of data collection (6 weeks to 6 months) is fairly long, and women’s opinions about their childbirth experience and future childbearing plans may change during this critical time, our study group could be heteronomous in their opinion.

- The low sample size is not included in the Limitations section.

We have added this information to the limitation section:

Therefore, although 733 women were included in the study, the final sample size of the longitudinal study is low

- Again, none of the appendices are included in the manuscript. 

We submitted four appendixes (A-D) with the manuscript in the system. We will inform the editor that you didn’t receive it.

Best wishes

Dorota Sys PhD